# Impact of Different Storage Methods on Bioactive Compounds in *Arthrospira platensis* Biomass

**DOI:** 10.3390/molecules24152810

**Published:** 2019-08-01

**Authors:** Teresa Papalia, Rossana Sidari, Maria Rosaria Panuccio

**Affiliations:** Department of Agricultural Science, “Mediterranea” University, Feo di Vito, 89124 Reggio Calabria, Italy

**Keywords:** *Artrospira platensis*, storage methods, bioactive compounds

## Abstract

*Arthrospira platensis* (spirulina) is considered a source of natural molecules with nutritional and health benefits. As the different storage forms can affect the quantity and quality of bioactive ingredients, the aim of the present work was to evaluate the effects of freezing, oven-drying and freeze-drying on chemical composition of spirulina biomass. Total proteins, photosynthetic pigments and antioxidants, were analyzed and compared to respective quantities in fresh biomass. The frozen sample exhibited the highest content of phycocyanin-C, phenols, and ascorbic acid, also respect to the fresh biomass. The highest total flavonoid amount was in the freeze-dried biomass. HPLC-DAD analysis of phenolic acids revealed the presence of the isoflavone genistein, known for its therapeutic role, in all the spirulina samples. The phosphomolybdenum method (TAC) and DPPH scavenging activity were applied to determine the antioxidant activity of different samples. The highest DPPH scavenging activity was detected in fresh and freeze-dried biomass and it was positively related to carotenoid content. A positive correlation indicated that carotenoids, chlorophyll, ascorbic acid and all phenolic compounds were the major contributors to the TAC activity in spirulina biomass. The results highlighted a different functional value of spirulina biomass, depending on the processing methods used for its storage.

## 1. Introduction

*Arthrospira (Spirulina) platensis* is a species of filamentous cyanobacteria that is included among the microalgae, a photosynthetic group consisting of eukaryotic and prokaryotic microorganisms. Spirulina has been used since ancient times both as a source food, for its protein (up to 70%, *w/w*) and vitamin (4%, *w/w*) content, and as an important source of valuable natural biologically active molecules, such as essential amino acids, minerals, long-chain polyunsaturated fatty acids and antioxidants [1]. Especially the phycobiliproteins, carotenoids and phenols contribute to antioxidant, immunomodulatory, and anti-inflammatory properties of this microalgae, playing protective roles on human health [2,3]. If chlorophyll and carotenoids are naturally occurring pigments present in all photosynthetic organisms, the phycocyanins are cyanobacteria pigments, involved in photosynthesis. They function by absorbing light in regions of low chlorophyll absorption and then increasing exciton transfer [4].

Light, temperature and nutrient availability are the main factors to be considered for the optimal growth of spirulina [5]. Many researchers studied the effects of manipulating growth conditions on the chemical composition of spirulina biomass in order to optimize the production of compounds with therapeutic and nutritional value [6,7]. Lack of essential nutrients, such as nitrogen, was employed to obtain lipids or carbohydrate-rich biomass [8,9]. Otherwise, the amounts of total carotenoids, chlorophylls-derived and phenolic compounds were positively related to the concentration of nitrogen in growth media [10,11]. These microalgae can tolerate extreme pH levels, excessive light, high heavy metals concentration and salinity thanks to the efficient mechanisms that protect cell homeostasis and the enhancement of antioxidants production in *A. platensis*, under different abiotic stress, has been widely reported [12,13,14].

There is an ever-growing interest for natural bioactive molecules which can be used in different industrial sectors. Other than microalgae [10,11], other microorganisms such as yeasts were found to be able to produce biologically active compounds useful for their antioxidant activity [15,16].

For commercialization purposes, spirulina storage and distribution processes are important steps to be considered to preserve the product, maintaining its nutritional and functional properties. In addition to the fresh biomass, over 75% of the products from spirulina are mainly in the dried form such as powders, tablets, capsules and also as extracts or processed in pasta, biscuits and other functional food products. To process spirulina, oven drying is one of the most conventional methods used, but the loss of water and the temperature during the drying processing can damage the heat-sensitive molecules and reduce the number and quality of the compounds compared to the original product. Freeze-drying is a technique that results in higher quality dehydrated products, preserving the raw material from thermal damage [17]. The low temperatures required and the solid state of water during freeze-drying protect the primary structure, but this is often considered an expensive and long drying process. On the other hand, during lyophilization, there might be a chance of a decrease in the antioxidant property caused by the degradation of certain bioactive molecules [18,19]. Freezing represents another methodology adopted by manufacturers for the conservation and distribution of spirulina biomass. Rapid freezing is used, with cooling temperatures from −30 to −40 °C, to avoid the risk that the increased water volume could damage the cell walls, causing lesions in the membranes and the leakage of solutes. By minimizing the size of the ice crystals through rapid cooling, the integrity of the cell membrane can be preserved even during thawing.

As the storage method can affect the chemical composition of the raw material, the different storage forms of spirulina may be endowed with different quantities and qualities of natural substances with nutritional and health benefits.

The purpose of this study was to identify the chemical compounds available in spirulina biomass differently stored. The results can contribute to defining the impact of the different storage processes on the nutritional and bioactive ingredients of spirulina and this is of interest to manufacturers and operators in the commercial distribution of quality spirulina-based products.

## 2. Results and Discussion

### 2.1. The Monitoring of A. platensis Cultivation

*A. platensis* cultivation was continued for 30 days. The growth curve showed a lag phase for the first three days, followed by an exponential phase until the 22nd day when the stationary phase began. A decline in the growth was evident after 27 days of culture. The cell productivity was 49.1 mg L^−1^day^−1^ and the highest value of biomass production was 1.12 g dry weight L^−1^. The culture was harvested after 20 days and the health status of culture was assessed by analyzing some biochemical parameters related to the carbon and nitrogen assimilation processes since both the adequate availability of nitrogen sources and HCO_3_ in the medium must be guaranteed to obtain an optimal growth [7,20] (Figure 1).

The chlorophyll fluorescence measurements give information on the photosynthesis process, and the ratio of variable fluorescence (Fv) over maximal fluorescence (Fm) is used as an indicator of the conversion efficiency of primary light energy and the maximal efficiency of PSII photochemistry. The Fv/Fm ratio of the culture was 0.52 and this result was in line with Fv/Fm values reported for *A. platensis* under normal growth conditions [21,22]. The nitrogen assimilation pathway is supported by carbon metabolism and all nitrogen sources are assimilated via the glutamine synthetase (GS)/glutamate synthase (GOGAT) cycle. In particular, the GOGAT enzyme is considered the link between both the nitrogen and carbon metabolisms [23,24]. At harvesting, the spirulina GOGAT activity was 20,625 U/mL and the residual nitrate in the medium was 15 mM, corresponding to 50% of the initial concentration, indicating an adequate nutritional supply in the medium (Figure 1).

### 2.2. Influence of Different Storage Methods on Photosynthetic Pigments of A. platensis

The different storage methods caused a decrease in the content of the photosynthetic pigments compared to fresh biomass, but at different extents depending on the conservation process (Table 1). Oven-drying caused the loss of 55% in the C-phycocyanin (C-PC) content and this could be a consequence of the C-PC sensitivity to heat treatment [25,26].

Otherwise, the content of C-PC was the highest in the frozen sample, also respect to the fresh biomass (Table 1). Probably, as the extraction of phycobiliproteins involves the cell rupture, the thawing favored the release of these proteins from within the cell. The quantities of phycoerythrin (PE) and allophycocyanin (APC) were lower compared to that of C-PC and their content decreased significantly in the various stored forms compared to fresh biomass. For the conservation of chlorophyll and carotenoid content, freezing was the least detrimental method with respect to the two drying techniques.

It is widely reported that phycocyanins and β-carotene are important active contributors to the health benefits of spirulina products [1,27,28,29]. The phycocyanins are cyanobacteria/microalgae photosynthetic pigments bonded to water-soluble proteins, known as phycobiliproteins, and analogous to light-collecting complexes of green plants. In the cells of *A. platensis*, C-PC can constitute up to 20% of the cell dry weight and it is the major protein pigment present with respect to APC and PE. These molecules are very sensitive to changes in light, temperature, pH, and extraction/purification solvents, and the different amounts of total phycocyanins (C-PC, APC, and PE) could explain different biological efficacies [30,31]. Phycocyanins from Spirulina are mostly consumed as natural edible and functional colorants due to their brilliant blue color with excellent antioxidant, anti-cancer, and anti-inflammatory activities. In addition to C-PC, APC possesses strong antioxidant activity, in terms of scavenging peroxyl radicals, whereas CPC is better at scavenging hydroxyl radicals [32].

Carotenoids are naturally occurring pigments in algae; spirulina cells mainly contain β-carotene and also cryptoxanthin and zeaxanthin [33]. Carotenoids are involved in light-harvesting reactions and in protection against photooxidation cell damage. The lipophilicity of these molecules, due to an extended system of conjugated double bonds, determine their major biochemical functions and their distribution in membranes and other lipophilic compartments. The β-carotene, in addition to its photoprotectant role, contributes significantly to the nutritional and therapeutic value of Spirulina for its pro-vitamin A action and the anti-inflammatory and antioxidants effects [27,34]. The low content of β-carotene in the spirulina dried (Table 2) can be due to the chemical deterioration of β-carotene through the drying and the storage stage, as this molecule is sensitive to oxygen because of its high degree of unsaturation [35].

### 2.3. Total Proteins and Antioxidant Molecules in A. platensis Samples

The amounts of soluble proteins, phenols and ascorbic acid (vitamin C), in the frozen sample, were higher than those in the fresh biomass (Table 2) and this may be a consequence of cellular membrane breakdown with better extraction of soluble molecules, caused by the low temperature. Freeze-drying did not lead to significant differences in the protein content, whereas oven-drying reduced the total protein by 35% with respect to the fresh biomass. In the oven-dried and in the lyophilized sample, the ascorbic acid content was lower with respect to that of both fresh and frozen biomass; conversely, the concentration of dehydroascorbic acid was significantly higher (Table 2). Ascorbic acid, due to its polar characteristics, is a hydrophilic antioxidant, and its degradation was found to be moisture- and temperature-dependent [36]. Additionally, a negative effect of oxygen on vitamin C retention was showed [37,38], thus, the area exposed to the drying conditions may be another detrimental factor and could explain the increase in the dehydroascorbic acid content, as a result of the oxidation of ascorbic acid.

### 2.4. Phytochemical Screening and Antioxidant Activity

Phenolic compounds are an important group of natural products acting as the primary antioxidant or free radical terminators and they have become very common constituents of the human diet because of their biological and therapeutic activities [39]. The phenol compounds in microalgae are structural components of cell walls and are involved in protection from UV radiation and in the cell-protective mechanism against abiotic and biotic stress and their content can be influenced by many factors (algal species, origin, growth conditions, and environmental variations as well) [40,41].

Recent studies demonstrated that even if the content of phenolic substances in microalgae was lower than that reported for plants, they may include novel phenolic compounds belonging to several classes of flavonoids, such as isoflavones, flavanones, flavonols [42,43]. A typical chromatographic profile of phenolic compounds in spirulina fresh biomass is reported in Figure 2.

In the fresh sample of *A. platensis*, *p*-hydroxybenzoic and the gallic acid were detected as representatives of the hydroxybenzoic acid class, and the *p*-coumaric, caffeic and ferulic acid as cinnamic acid derivatives (Table 3). Similar results on the characterization of phenolic acids in spirulina were reported by Klejdus et al [44].

These phenols were not present in all the extracts, except for the gallic acid, a phenolic acid widely distributed and well known for its neuroprotective actions [45].

Caffeic acid, considered one of the most effective antioxidants among hydroxicinnamic acids, was only present in fresh and frozen biomass. The benzoic and cinnamic acid derivatives are basic precursor species for the synthesis of polyphenolic compounds that are generally subdivided into 2 large groups: flavonoids and nonflavonoid. The common characteristic of flavonoids is the basic 15 carbon structure arranged in 3 rings and the classes differ primarily for the state of oxidation of central carbon ring. The highest flavonoids concentration was found in freeze-dried biomass, with a content of 40% more than that of the fresh biomass (Table 2) and this result agrees with previous reports showing a positive effect of the freeze-drying process on the level of flavonoids, facilitating their extraction [42,46]. Qualitative analysis showed that all samples, except for the oven-dried ones, contained the two flavonols, quercetin and kaempferol, and the flavanol catechin was undetectable in the freeze-dried sample (Table 3).

The antioxidant potential of flavonoids is positively related to the number and location of free-OH groups on their skeleton [43,47], as flavanols and flavonols of *A. platensis* have multiple OH groups, they can effectively contribute to the radical scavenging potential of this cyanobacteria. The isoflavone genistein was found in all samples (Table 3). This isoflavone is not widespread in food and is mainly present in soy and its products, and the poor water solubility is the major limitation to its availability [48].

Genistein is an example of a phytoestrogenic compound due to its chemical structure similar to estradiol and its estrogen receptor binding. Several experimental and clinical investigations suggest an important therapeutic role of genistein on different types of cancer because it can promote cancer cell death by inducing apoptosis and other cytotoxic processes. Furthermore, McCarty et al. (2009) suggested that genistein and phycocyanobilin might have a considerable potential for a joint positive effect on the prevention of some hepatic diseases [49,50,51].

The phosphomolybdenum method and the antiradical scavenging activity using 1,1-Diphenyl-2-Picrylhydrazyl (DPPH) were applied to determine the antioxidant activity in spirulina samples. DPPH can accept an electron or hydrogen radical and it is used to measure the overall antioxidant capacity for both hydrophilic and lipophilic antioxidants [52]. The highest DPPH scavenging activity was detected in fresh and freeze-dried biomass (Figure 3) A positive correlation was observed between DPPH radical scavenging activity and the carotenoid content in fresh, oven-dried and lyophilized samples (Table 4), confirming the significant contribution of carotenoids to some antioxidant properties of microalgae, as previously reported [53]. Additionally, in freeze-dried biomass, C-PC, chlorophyll, total flavonoids and ascorbate content exhibited a positive correlation with the DPPH assay. The DPPH results in the frozen spirulina and even more in the oven-dried one were significantly lower with respect to fresh biomass (Figure 3).

Total Antioxidant Capacity (TAC), by using the phosphomolybdenum reduction assay, was expressed as α-tocopherol, for the evaluation of fat-soluble antioxidant capacity. The α-tocopherol (vitamin E) is a compound belonging to the group of lipid-soluble antioxidants. It is located in the plastid or thylakoid membranes of plants, algae and some cyanobacteria with the main role of protecting these organisms from the photosynthesis-derived reactive oxygen species (ROS) [54]. In the oven-dried sample, TAC activity was significantly reduced compared to that of fresh biomass, while in frozen and lyophilized spirulina, the decrease in the α-tocopherol content was smaller, and it was 6% and 14%, respectively. The positive correlation data (Table 4) showed that carotenoids, chlorophyll, ascorbic acid and all phenolic compounds were the major contributors to the TAC activity in spirulina samples.

## 3. Materials and Methods

### 3.1. Chemicals

HPLC-grade solvents were purchased from Merck (Darmstadt, Germany). Water used throughout the study was purified on a Milli-Q system from Millipore (Bedford, MA, USA). All the available standard reference compounds were purchased from Extrasynthese (Genay, France). All the other reagents and chemicals used in this study were of analytical grade and were purchased from Sigma (Sigma-Aldrich GmbH, Sternheim, Germany).

### 3.2. Culture Conditions for A. platensis and Application of Drying Methods

*A. platensis.* was provided by the CRIAcq Research Center of the University of Naples. Cultivation was conducted for a period of 30 days in 500 mL Erlenmeyer flasks containing 200 mL of Zarrouk’s medium [55] (pH 9.0) at 30 ± 2 °C, under the illumination of 3500 ± 100 lux using cool white fluorescent lamps, with a photoperiod light/dark of 12:12 h and thrice daily mechanical shaking. The experiment was carried out in triplicates and the algal growth was monitored by measurements of optical density at 550 nm (g L^−1^) by using the Spectrophotometer UV-1800 (Shimatzu).

Cell productivity was calculated from the equation P = (Xi−X0)/ti, where *P* = productivity (mg L^−1^day^−1^), *X*0 = initial biomass density (mg L^−1^), *Xi* = biomass density at ti (mg L^−1^) and ti *=* time interval (h) between X0 and Xi [56].

After 20 days, during the exponential phase of growth, the spirulina culture was harvested to detect the photosynthetic efficiency. For this, the parameter related to the maximum quantum yield of PSII photochemistry (Fv/Fm) was evaluated by using an Imaging PAM Fluorometer (Walz, Effeltrich, Germany) [57].

For glutamate synthase activity determination (GOGAT EC 1.4.7.1), the cell suspension was filtered under a vacuum (Millipore filter membrane 0.45 µm) and added in a 20 mM phosphate buffer (pH 7.5), ratio 1:3 (*w/v*), with 0.5 mM EDTA and 0.1 mM DTT. Then, the sample was sonicated at 4 °C for 3 min. (10 s on, 20 s off) at 18 KHz, using a Transsonic 460/H Elma cell disrupter and centrifuged at 10,000 rpm for 10 min at 4 °C. The supernatant was used for the GOGAT assay [58]. One unit of the specific activity of GOGAT was expressed as the amount of enzyme needed to produce 1 nmol of glutamate per milligram of protein at 30 °C.

The culture of *A. platensis.* was harvested at the end of 20th day. The algal suspension was filtered under vacuum (0.45 µm) and washed several times with distilled water to remove soluble salts. The recovered biomass was studied as fresh, frozen (at −22 °C) and dried cells by the freeze-drying (lyophilization) and oven-drying methods. The former method was carried out using a lyophilizer (Labconco Free Zone 2.5 Liter Freeze Dry System) while in the latter method, the biomass was spread in a thin layer before being processed in an oven at 50 °C for 6 h. The different samples were analyzed and compared to fresh biomass (control). All values measured are corrected for dry basis. The dry weight was determined after drying at 105 °C, until constant weight. The difference between the initial and final weight was the water percentage of the spirulina biomass.

### 3.3. Protein Determination

Protein extraction was carried out according to the Payne and Stewart method [59] with some modifications. Algal biomass (0.5 g) was added to 0.5 N NaOH (6 mL), sonicated for 2 min and then extracted in a water bath for 20 min at 80 °C. The protein content (mg/g dry weight) was quantified by UV-Visible electronic absorption at 595 nm using the Bradford method [60].

### 3.4. Phycobiliproteins

The biomass fresh, frozen, oven-dried and freeze-dried were extracted with a phosphate buffer (0.05 M, pH 6.7) ratio 1:3 (*w/v*), by repeatedly freezing and thawing three times. Afterward, the samples were centrifuged for 15 min at 10,000 rpm and the supernatants were analyzed by spectrophotometer (Shimatzu UV-1800). The absorbance was read at 562 nm (A_562_), 615 nm (A_615_), and 652 nm (A_652_) against a blank and the concentrations of C-phycocyanin C (C-PC), allophycocyanin (APC), and phycoerythrin (PE) were calculated by using the following equations [61]:

C-Phycocyanin (C-PC) = [OD_615_ − 0.474(OD_652_)]/5.34

Allophycocyanin (APC) = [OD_652_ − 0.208(OD_615_)]/5.09

Phycoerythrin (PE) = [OD_562_ − 2.41(PC) − 0.849(APC)]/9.62

### 3.5. Chlorophyll and Carotenoids

The photosynthetic pigments were extracted from different spirulina samples (50 mg) by using 5 mL 90% acetone in the dark, for 24 h at 4 °C. Then, the sample was centrifuged for 15 min at 5000 rpm and the supernatant was collected. The absorbance for chlorophyll a and carotenoids was measured at 470 nm (A_470_), 649 nm (A_649_), 645 nm (A_645_), and 665 nm (A_665_) against 90% acetone as blank, by a Shimatzu UV/VIS spectrophotometer and the concentrations of pigments were calculated using the Lichtenthaler’s equations [39].

### 3.6. Ascorbic and Dehydroascorbic Acid

Algal biomass (0.5 g) was extracted with a solution of metaphosphoric acid (5%), centrifuged at 18,000 rpm for 10 min and the supernatant was used for the determination of ascorbic acid (ASC) and dehydroascorbic acid (DHA) [62].

### 3.7. Total Phenol and Total Flavonoid Analysis

One gram of differently stored algal biomass was suspended in ethanol (5 mL), sonicated to disrupt cells and homogenized for 3 min at 4 °C. The homogenate was centrifuged at 2000 rpm for 15 min at 4°C, the resulting supernatant was filtered through Millipore filters (0.45 µm pore size). The filtrate was evaporated to dryness and resuspended in a final volume of 1.0 mL with ethanol [11]. 

The total phenol content was determined using the Folin-Ciocalteu assay [63]. Ethanol extracts (0.04 mL) were added to 1.6 mL of H_2_O and 0.1 mL of Folin-Ciocalteu reagent and incubated at 25 °C for 10 min. Afterward, 0.6 mL of a 7.5% solution of Na_2_CO_3_ was added to each sample and left at 40 °C for 20 min in a water bath, with intermittent shaking. The absorbance of the samples was recorded at 760 nm. The calibration curve was performed with gallic acid and the results were expressed as mg of gallic acid equivalents per g of dry weight.

The total flavonoid content was spectrophotometrically determined by the aluminum chloride method based on the formation of complex flavonoid—aluminum [64]. A total of 0.5 milliliters of ethanolic extracts were mixed with 1 mL of AlCl_3_ methanolic solution (2%, *w/v*). After incubation at room temperature for 15 min, the absorbance of the reaction mixture was measured at 430 nm. The contents of TFC were estimated from the standard calibration curve quercetin (mg/g dry weight).

### 3.8. Antioxidant Activity Assays

The compound 2,2-diphenyl-1-picrylhydrazyl radical (DPPH^•^) scavenging assay was performed according to Blois [65]. DPPH^•^ concentration in the cuvette was chosen to give absorbance values of ~1.0. The reaction mixtures were composed of 10 μL of ethanolic extract, 700 μL DPPH^•^ and ethanol up to the final of 1.0 mL. A blank, without-ethanol-extract, was prepared for each sample. The change in absorbance of the violet solution was recorded at 517 nm after 30 min of incubation at 37 °C. The inhibition (%) of radical scavenging activity was calculated by the following equation:(1)$$Inhibition(\%)=\frac{A_{O}-A_{S}}{A_{O}}\times 100$$
where *A_O_* is the absorbance of the control and *A_S_* is the absorbance of the sample after incubation.

The total antioxidant capacity (TAC) was measured by the method of Prieto [66]. Algal biomass was homogenized in 50% methanol and was centrifuged after at 4 °C at 10,000 rpm for 20 min; the supernatant was used and the absorbance was measured at 546 nm to express the total antioxidant capacity as μg α-tocopherol/g dry weight.

### 3.9. Phenolic Profile Identification

One gram of algal biomass was extracted using 5 mL of ethanol with constant stirring for 24 h at room temperature. The extract was filtered and the process was repeated thrice. The extracts were combined, filtered through a 0.45 μm PTFE membrane and the volume was made up to 1 mL using the same solvent before storing. The identification of flavonoids was performed by HPLC according to the method reported by Seal [67]. The instrument was a Dionex Ultimate 3000 liquid chromatography (Germany) with a diode array as the detector (DAD 3000) and the Chromeleon 7 system manager as the data processor. The separation was achieved by a reversed-phase Acclaim TM 120 C18 column (5 m particle size, i.d. 4.6 × 250 mm).

The mobile phase was a 1% acetic acid solution (Solvent A) and acetonitrile (Solvent B), the flow rate was adjusted to 0.7 mL/min, the column was thermostatically controlled at 28 °C and the injection volume was kept at 20 μL. The gradient elution was 10 to 40% B in a linear fashion for 28 min, from 40 to 60% B for 39 min, from 60 to 90% B for 50 min. HPLC chromatograms were detected using a photodiode array UV detector at three different wavelengths (272, 280 and 310 nm) according to the maximum absorption of the analyzed compounds. Each compound was identified by its retention time (Rt) and by using the corresponding standard under the same conditions. 

### 3.10. Statistical Analysis

The values of the data are expressed as means ± standard error. One-way analysis of variance (ANOVA) has been performed on the obtained results. Tukey’s test was run to check the significance of the difference between the samples and the respective controls. A *p* < 0.05 value indicates significantly statistical difference. All analyses were conducted using SYSTAT 13 for Windows.

## 4. Conclusions

*A. platensis* is considered an important nutritional supplement for the abundance of its natural bioactive compounds. Especially carotenoids, tocopherols, ascorbic acid and chlorophyll derivatives are key components of the antioxidant cellular network, which is due to the synergistic contribution of all the compounds rather than to the efficacy of the single antioxidant. The results of this study show that the spirulina has different functional value depending on the processing methods used for its storage. Therefore, it is difficult to indicate the best storage method for spirulina biomass that can be usefully applied for all commercial applications, such as medicine, food industry and agriculture. Moreover, the treatment of the biomass can be responsible for some significant losses of the qualitative properties. These aspects are important to take into account at the production sites as well as throughout the commercialization of spirulina products to preserve the quantity and quality of natural substances unaltered with nutritional and health benefits.

## Figures and Tables

**Figure 1 molecules-24-02810-f001:**
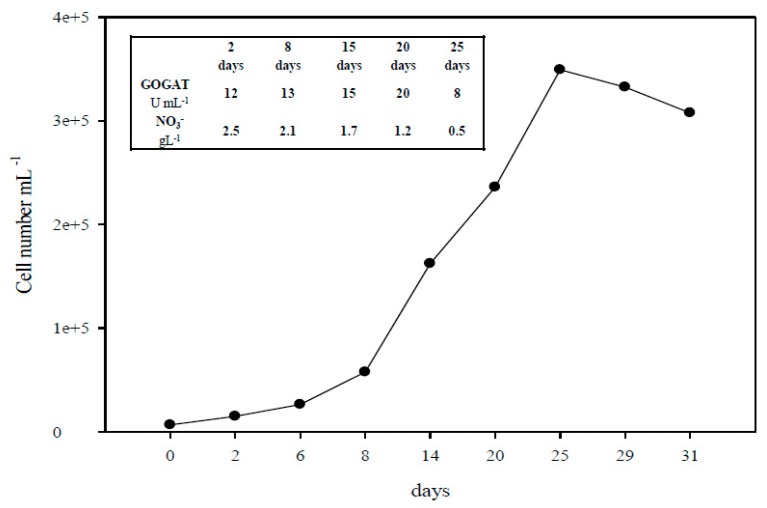
Cell growth curve, Glutamate Synthase (GOGAT) activity and nitrate depletion in the medium during the cultivation period.

**Figure 2 molecules-24-02810-f002:**
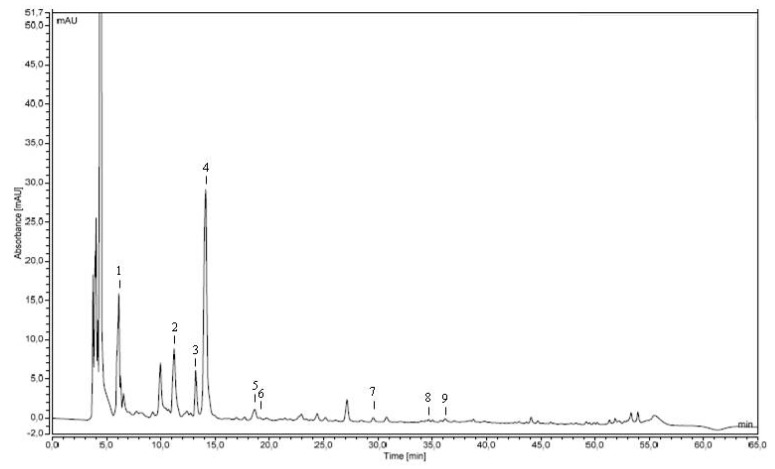
The representative chromatogram of phenolic compounds extracted from the fresh biomass of *Spirulina platensis*: absorbance at 272 nm. Peak identifications were performed by retention time and UV spectra against commercially available reference compounds. Peaks: gallic acid (**1**) catechin (**2**), caffeic acid (**3**), *p*-hydroxybenzoic acid (**4**), *p*-coumaric acid (**5**), ferulic acid (**6**) quercetin (**7**), genistein (**8**) and kaempferol (**9**).

**Figure 3 molecules-24-02810-f003:**
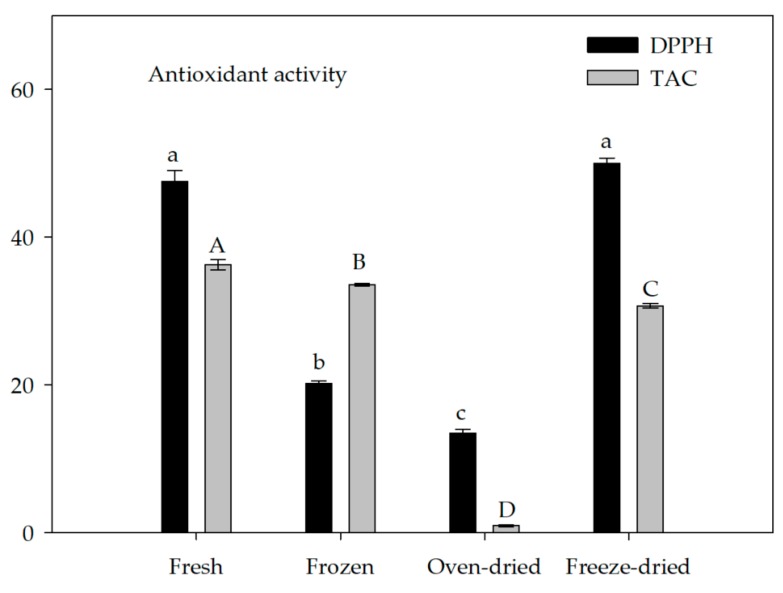
The 1,1-Diphenyl-2-Picrylhydrazyl radical scavenging activity (DPPH, % inhibition) and Total Antioxidant Activity (TAC, µg α-tocopherol g D.W.^−1^). Different letters indicate, within the same assay, significant differences (*p* ≤ 0.05) among *A. platensis* samples.

**Table 1 molecules-24-02810-t001:** The pigments content in *A. platensis* biomass fresh and differently processed.

Biomolecules	*A. platensis* Fresh	*A. platensis* Frozen	*A. platensis* Oven-Dried	*A. platensis* Freeze-Dried
**C-Phycocyanin (mg g D.W.^−^^1^)**	44.81 ± 0.35b	50.21 ± 0.74a	18.88 ± 0.35d	38.03 ± 0.50c
**Allophycocyanin (mg g D.W.^−^^1^)**	18.90 ± 1.03a	4 ± 0.09b	4.52 ± 0.51b	5.48 ± 0.16b
P**hycoerythrin (mg g D.W.^−^^1^)**	5.28 ± 0.16a	0.99 ± 0.25c	1.47 ± 0.26c	2.17 ± 0.11b
**Chlorophyll a (mg g D.W.^−^^1^)**	12.45 ± 0.04a	9.95 ± 0.09b	1.47 ± 0.03d	6.44 ± 0.03c
**Carotenoids (mg g D.W.^−^^1^)**	3.82 ± 0.15a	3.31 ± 0.05b	0.90 ± 0.001d	2.22 ± 0.004c

Data are expressed as the mean ± s.e. (standard error). Different letters, in the same row, indicate significant differences (*p* ≤ 0.05).

**Table 2 molecules-24-02810-t002:** The total proteins and antioxidant molecules in *A. platensis* biomass fresh and differently processed.

Biomolecules	*A. platensis* Fresh	*A. platensis* Frozen	*A. platensis* Oven-Dried	*A. platensis* Freeze-Dried
**Total Proteins** **(mg g D.W. ^−^^1^)**	188.60 ± 13.55b	283.96 ± 11.79a	122.73 ± 2.53c	167.09 ± 4.35b
**Ascorbic acid** **(mg g D.W.^−^^1^)**	1678.292.62b	3149.54 ± 7.99a	354.79 ± 0.93d	1403.9 ± 11.49c
**Dehydroascorbic acid** **(mg g D.W.^−^^1^)**	1998.99 ± 7.01c	1362.93 ± 22.78d	3296.69 ± 15.56b	4660.74 ± 34.56a
**Total Phenols** **(mg g D.W.^−^^1^)**	15.77 ± 1.10b	22.65 ± 0.46a	12.14 ± 1.84c	11.91 ± 0.28c
**Total Flavonoids** **(mg g D.W.^−^^1^)**	20.82 ± 0.08b	8.04 ± 0.29c	4.31 ± 0.11d	30.92 0.17a

Data are expressed as the mean ± s.e. (standard error). Different letters, in the same row, indicate significant differences (*p* ≤ 0.05).

**Table 3 molecules-24-02810-t003:** The phenolic compounds in *A. platensis* samples.

Compounds	Rt	*A. platensis* Fresh	*A. platensis* Oven-Dried	*A. platensis* Frozen	*A. platensis* Freeze-Dried
Gallic acid	6.11	+	+	+	+
Catechin	11.28	+	+	+	-
Caffeic acid	13.22	+	-	+	-
*p*-Hydroxybenzoic acid	14.13	+	+	+	-
*p*-Cumaric acid	18.69	+	+	-	+
Ferulic acid	18.81	+	+	-	+
Quercetin	29.59	+	-	+	+
Genistein	34.95	+	+	+	+
Kaempferol	36.67	+	+	+	+

Rt, retention time; +, compound detected; -, compound not detected.

**Table 4 molecules-24-02810-t004:** The correlation among the antioxidant compounds and antioxidant activities of *A. platensis* are differently processed.

	Biomass	C-PC	CAROT	CHL	PHEN	FLAVON	ASC
**DPPH**	**Fresh**						
	r	−0.994	0.879	−0.736	0.546	0.147	−0.087
	R^2^	0.989	0.77	0.536	n.s.	n.s.	n.s.
	**Frozen**						
	r	−0.099	−0.13	−0.805	−1.00	−0.528	−0.926
	R^2^	n.s.	n.s.	0.648	0.999	n.s.	0.857
	**Oven-dried**						
	r	0.624	0.776	−0.928	−0.586	−0.863	0.541
	R^2^	n.s.	0.603	0.861	n.s.	0.745	n.s.
	**Freeze-dried**						
	r	0.961	0.926	0.895	0.676	0.977	0.709
	R^2^	0.924	0.858	0.801	n.s.	0.954	0.502
**TAC**	**Fresh**						
	r	0.207	0.385	−0.604	0.778	0.969	−0.982
	R^2^	n.s.	n.s.	n.s.	0.606	0.939	0.965
	**Frozen**						
	r	0.084	0.116	0.796	1.00	0.516	0.920
	R^2^	n.s.	n.s.	0.634	1.00	n.s.	0.847
	**Oven-dried**						
	r	0.571	−0.999	0.516	0.956	0.993	0.938
	R^2^	n.s.	0.999	n.s.	0.913	0.986	0.88
	**Freeze-dried**						
	r	−0.934	0.899	0.863	0.627	0.960	0.66
	R^2^	0.872	0.808	0.745	n.s.	0.922	n.s.

C-PC, phycocyanin C; CAROT, carotenoids; CHL, chlorophyll a; PHEN, total phenols; FLAVON, total flavonoids; ASC, ascorbic acid.

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
