# Peer review of "Impact of Different Storage Methods on Bioactive Compounds in Arthrospira platensis Biomass"

_molecules, 2019, doi:10.3390/molecules24152810_

Round 1
Reviewer 1 Report
Referees Comments
on the manuscript entitled "Impact of different storage methods on bioactive compounds in Arthrospira platensis biomass” for Molecules
There is an ever-increasing search for nontoxic original innovative molecules which can be produced by microorganisms, algae at a relatively large scale and which can find wide application in medicine, food industry and agriculture, that is, where there are high requirements to the purity of such compounds, to their stereo- and isomeric composition, and to ability to be metabolized in the cells to carbon dioxide and water without accumulation.
In this connection the subject of manuscript is of a high interest and in the scope of Molecules and could be considered for publication in this journal but the revision of the manuscript is required:
1. More recently, it was discovered that unconventional yeasts Yarrowia lipolytica are capable to produce the biologically active pharmaceutically pure molecules. Morgunov et al. (2018) studied the effect of biologically active threo-Ds-isocitric acid on oxidative stress in the infusorian Paramecium caudatum stressed by hydrogen peroxide and salts of some heavy metals (Cu, Pb, Zn, and Cd); isocitric acid was found to be a more active antioxidant than ascorbic acid. Also Morgunov et al. (2019) observed that the monopotassium salt of isocitric acid isolated from the culture liquid and purified to 99.9% was found to remove neurointoxication, to restore memory and to improve the learning of laboratory rats intoxicated with lead and molybdenum salts. Taking into account the fact that the neurotoxic effect of heavy metals is mainly determined by oxidative stress, the aforementioned favorable action of isocitric acid on the intoxicated rats can be explained by its antioxidant activity among other pharmacological effects.
Also Morgunov et al. (2017) reported the biological activity of natural molecules against phytopathogens. It was found that succinic acid produced by microbiological synthesis possesses the bactericidal, fungicidal, and nematodocidal activities (Kamzolova et al. 2014).
Please to include these high-relevant investigations in the manuscript:
Morgunov, I.G.; Karpukhina, O.V.; Kamzolova, S.V.; Samoilenko, V.A.; Inozemtsev, A.N. Investigation of the effect of biologically active threo-Ds-isocitric acid on oxidative stress in Paramecium caudatum. Prep. Biochem. Biotechnol, 2018, 48, 1-5. doi: 10.1080/10826068.2017.1381622
Morgunov, I.G.; Kamzolova, S.V.; Karpukhina, O.V.; Bokieva, S.B.; Inozemtsev, A.N. Biosynthesis of isocitric acid in repeated-batch culture and testing of its stress-protective activity. Appl. Microbiol. Biotechnol. 2019, 103, 3549-3558. doi: 10.1007/s00253-019-09729-8.
2. Page 2, lines 50-51 - The purpose of the work is recommended to write in a traditional manner, with a new line.
3. Page 2, lines 52-58 - The description of the sequence of work and the results are recommended to be removed from the Introduction.
4. Page 2, lines 61-79 – Please to include a figure with the dynamics of growth of A. platensis, consumption of carbon and nitrogen sources and other relevant parameters.
5. Page 3 – Table 1 must be placed at the top of the page, immediately after the description of the results. In Table 1 to write the name of the first column.
6. Page 3, line 95 – To deselect underscore.
7. Page 3, line 112 – To change (Tab.2 ) to (Table 2).
8. Page 4 - In Table 2 to write the name of the first column.
9. Page 4 - Table 2 should be moved to section 2.3.Total proteins and antioxidant molecules in A. platensis samples, because in this section the results of Table 2 were described.
10. Page 5 - Table 3 must be placed at the top of the page, immediately after the description of the results. Please to correct the misprint for the “retation time”.
11. Page 7 - Move the Legend under the Figure 1.
Author Response
Response to Reviewer 1 Comments
Point 1: There is an ever-increasing search for nontoxic original innovative molecules which can be produced by microorganisms, algae at a relatively large scale and which can find wide application in medicine, food industry and agriculture, that is, where there are high requirements to the purity of such compounds, to their stereo- and isomeric composition, and to ability to be metabolized in the cells to carbon dioxide and water without accumulation. In this connection the subject of manuscript is of a high interest and in the scope of Molecules and could be considered for publication in this journal but the revision of the manuscript is required: 1. More recently, it was discovered that unconventional yeasts Yarrowia lipolytica are capable to produce the biologically active pharmaceutically pure molecules. Morgunov et al. (2018) studied the effect of biologically active threo-Ds-isocitric acid on oxidative stress in the infusorian Paramecium caudatum stressed by hydrogen peroxide and salts of some heavy metals (Cu, Pb, Zn, and Cd); isocitric acid was found to be a more active antioxidant than ascorbic acid. Also Morgunov et al. (2019) observed that the monopotassium salt of isocitric acid isolated from the culture liquid and purified to 99.9% was found to remove neurointoxication, to restore memory and to improve the learning of laboratory rats intoxicated with lead and molybdenum salts. Taking into account the fact that the neurotoxic effect of heavy metals is mainly determined by oxidative stress, the aforementioned favorable action of isocitric acid on the intoxicated rats can be explained by its antioxidant activity among other pharmacological effects. Also Morgunov et al. (2017) reported the biological activity of natural molecules against phytopathogens. It was found that succinic acid produced by microbiological synthesis possesses the bactericidal, fungicidal, and nematodocidal activities (Kamzolova et al. 2014). Please to include these high-relevant investigations in the manuscript: Morgunov, I.G.; Karpukhina, O.V.; Kamzolova, S.V.; Samoilenko, V.A.; Inozemtsev, A.N. Investigation of the effect of biologically active threo-Ds-isocitric acid on oxidative stress in Paramecium caudatum. Prep. Biochem. Biotechnol, 2018, 48, 1-5. doi: 10.1080/10826068.2017.1381622
Morgunov, I.G.; Kamzolova, S.V.; Karpukhina, O.V.; Bokieva, S.B.; Inozemtsev, A.N. Biosynthesis of isocitric acid in repeated-batch culture and testing of its stress-protective activity. Appl. Microbiol. Biotechnol. 2019, 103, 3549-3558. doi: 10.1007/s00253-019-09729-8.
Response 1: The introduction was implemented and references have been added.
Point 2: Page 2, lines 50-51 - The purpose of the work is recommended to write in a traditional manner, with a new line.
Response 2: The aim has been moved in a new line.
Point 3: Page 2, lines 52-58 - The description of the sequence of work and the results are recommended to be removed from the Introduction.
Response 3: Lines 52-58 have been removed.
Point 4: Page 2, lines 61-79 – Please to include a figure with the dynamics of growth of A. platensis, consumption of carbon and nitrogen sources and other relevant parameters.
Response 4: The required figure was included.
Point 5: Page 3 – Table 1 must be placed at the top of the page, immediately after the description of the results. In Table 1 to write the name of the first column.
Response 5: Table was modified and moved immediately after the description of the results
Point 6: Page 3, line 95 – To deselect underscore.
Response 6: The changes were made .
Point 7: Page 3, line 112 – To change (Tab.2 ) to (Table 2).
Response 7: Modification has been done.
Point 8: Page 4 - In Table 2 to write the name of the first column.
Response 8: Table 2 was corrected.
Point 9: Page 4 - Table 2 should be moved to section 2.3.Total proteins and antioxidant molecules in A. platensis samples, because in this section the results of Table 2 were described.
Response 9: Table 2 was moved to section 2.3.
Point 10: Page 5 - Table 3 must be placed at the top of the page, immediately after the description of the results. Please to correct the misprint for the “retation time”.
Response 10 : Table 3 was modified and moved.
Point 11: Page 7 - Move the Legend under the Figure 1.
Response 11: The legend was moved.
Reviewer 2 Report
The improvement on explanation about the Table 3 might be recommended. Such as the meaning of symbol ‘X’ cannot be easily understandable for the readers.
If possible, a figure of one of the typical chromatographic profile should be added. The identification of each compound is not unclear.
In addition, the suppliers of standard compounds of polyphenols should be described in the Materials and Methods part.
Author Response
Response to Reviewer 2 Comments
Point 1: The improvement on explanation about the Table 3 might be recommended. Such as the meaning of symbol ‘X’ cannot be easily understandable for the readers.
Response 1: The symbol ‘X’ was changed to make understandable the meaning for the readers.
Point 2: If possible, a figure of one of the typical chromatographic profile should be added. The identification of each compound is not unclear.
Response 2: The figure has been added.
Point 3: In addition, the suppliers of standard compounds of polyphenols should be described in the Materials and Methods part.
Response 3: The suppliers of all chemicals used are reported in section 3.1 of Materials and Methods.
Reviewer 3 Report
Discussion:
- Page 5: Include the quantification results in Table 3 and validation parameters
- Page 5 line 180: What is PC-P. It would be C-PC?
- Page 5 line 181: change the word “dried” to “oven dried”. Observe it along the discussion (like page 7 line192)
- Page 6, Table 4: Change “PC-P” to “C-PC”. CAROT description is missing
- Page 6, Figure 1: Y axis title is missing/ Change the word “dried” to “oven-dried” and “lyophilized” to “freeze-dried”.
Materials and Methods:
- To improve your work I suggest the use of refined techniques for individual components quantification (carotenoids, vitamin C, chlorophyll).
- For antioxidant measures, use ROS scavenging (peroxyl radical scavenging – ORAC; Hydroxyl radical; HOCl scavenging; superoxide scavenging, for example)
Conclusions:
- The conclusion is very general. What is the best storage method to spirulina biomass and its components?
Author Response
Response to Reviewer 3 Comments
Point 1: Page 5: Include the quantification results in Table 3 and validation parameters
Response 1: We performed only a qualitative analysis of phenolic compounds using a validated method reported in literature.
Point 2: Page 5 line 180: What is PC-P. It would be C-PC?
Response 2: The word has been corrected.
Point 3: Page 5 line 181: change the word “dried” to “oven dried”. Observe it along the discussion (like page 7 line192)
Response 3: The word has been changed along the discussion.
Point 4: Page 6, Table 4: Change “PC-P” to “C-PC”. CAROT description is missing
Response 4: Table 4 was corrected.
Point 5: Page 6, Figure 1: Y axis title is missing/ Change the word “dried” to “oven-dried” and “lyophilized” to “freeze-dried”.
Response 5 : The figure has been modified. DPPH and TAC are expressed with different units, as reported in the legend of the figure.
Point 6: Materials and Methods: To improve your work I suggest the use of refined techniques for individual components quantification (carotenoids, vitamin C, chlorophyll).
Response 6: In our opinion, the results of spectrophotometric analysis can provide satisfactory results considering the aim of this study.
Point 7: For antioxidant measures, use ROS scavenging (peroxyl radical scavenging – ORAC; Hydroxyl radical; HOCl scavenging; superoxide scavenging, for example)
Response 7: Our purpose at this step of investigation was to determine the differences in the total antioxidant activity of spirulina biomass differently stored. Thanks for your suggestion, it will be useful for future work.
Point 8: Conclusions: The conclusion is very general. What is the best storage method to spirulina biomass and its components?
Response 8: The conclusions have been improved.